# Chemokines as Regulators of Neutrophils: Focus on Tumors, Therapeutic Targeting, and Immunotherapy

**DOI:** 10.3390/cancers14030680

**Published:** 2022-01-28

**Authors:** Raffaella Bonecchi, Alberto Mantovani, Sebastien Jaillon

**Affiliations:** 1Department of Biomedical Sciences, Humanitas University, Via Rita Levi Montalcini 4, 20090 Pieve Emanuele, MI, Italy; alberto.mantovani@hunimed.eu (A.M.); sebastien.jaillon@hunimed.eu (S.J.); 2IRCCS Humanitas Research Hospital, via Manzoni 56, 20089 Rozzano, MI, Italy; 3The William Harvey Research Institute, Queen Mary University of London, London EC1M 6BQ, UK

**Keywords:** neutrophils, chemokines, chemokine receptors, inflammation and cancer, immunotherapy

## Abstract

**Simple Summary:**

Neutrophils are the main leukocyte subset present in human blood and play a fundamental role in the defense against infections. Neutrophils are also an important component of the tumor stroma because they are recruited by selected chemokines produced by both cancer cells and other cells of the stroma. Even if their presence has been mostly associated with a bad prognosis, tumor-associated neutrophils are present in different maturation and activation states and can exert both protumor and antitumor activities. In addition, it is now emerging that chemokines not only induce neutrophil directional migration but also have an important role in their activation and maturation. For these reasons, chemokines and chemokine receptors are now considered targets to improve the antitumoral function of neutrophils in cancer immunotherapy.

**Abstract:**

Neutrophils are an important component of the tumor microenvironment, and their infiltration has been associated with a poor prognosis for most human tumors. However, neutrophils have been shown to be endowed with both protumor and antitumor activities, reflecting their heterogeneity and plasticity in cancer. A growing body of studies has demonstrated that chemokines and chemokine receptors, which are fundamental regulators of neutrophils trafficking, can affect neutrophil maturation and effector functions. Here, we review human and mouse data suggesting that targeting chemokines or chemokine receptors can modulate neutrophil activity and improve their antitumor properties and the efficiency of immunotherapy.

## 1. Introduction

Neutrophils are the most abundant circulating leukocytes in humans, accounting for 50–70% of all circulating white blood cells, and represent the first line of defense against bacterial and fungal pathogens. Neutrophils are fundamental for our immune response not only for their antimicrobial functions but also for the orchestration and regulation of the innate and adaptive immune responses [1]. In addition, a growing number of studies have demonstrated the heterogeneity and plasticity of neutrophils in different contexts, including tissue homeostasis and pathological conditions [2].

In addition to cancer cells, the tumor microenvironment (TME) is composed of extracellular matrix, stromal cells, and inflammatory cells. Neutrophils are an important component of the TME, and in the majority of human tumors, their infiltration has been associated with a poor prognosis [3]. However, the role of neutrophils in cancer must be reconsidered due to their plasticity and heterogeneity [4]. Indeed, neutrophils can have both protumor and antitumor activities reflecting their different activation and differentiation states [2,4].

Neutrophil recruitment in the TME is mainly driven by chemokines. Here, we review the current understanding of the heterogeneity and role played by neutrophils in cancer focusing our attention on chemokines and chemokine receptors involved in neutrophil development, maturation, recruitment, and activation states and their possible targeting for immunotherapy.

## 2. Neutrophil Development

Neutrophils are produced in the bone marrow (BM) from hematopoietic stem cells (HSC) that differentiate successively into lymphoid-primed multipotent progenitors (LMPPs) or common myeloid progenitors (CMPs), granulocyte monocyte progenitors (GMPs), and mature neutrophils equipped with granules. This process is mainly regulated by the expression of granulocyte colony-stimulating factor (G-CSF) and granulocyte-macrophage colony-stimulating factor (GM-CSF) [5,6,7]. Accordingly, severe neutropenia has been observed in G-CSF^−/−^ and GM-CSF^−/−^ mice [8]. Due to their limited life span, their presence in the blood requires the production of up to 2 × 10^11^ neutrophils per day [6].

Using mass cytometry by time of flight (CyTOF) and single-cell RNA sequencing (scRNA-seq), recent studies have identified different subsets of immature neutrophils within the BM [9,10,11,12]. In particular, committed proliferative neutrophil precursors (referred to as preNeu), characterized as a proliferative population of Gr1^+^ CD11b^+^ CXCR4^hi^ CD117^int^ CXCR2^−^ cells which differentiate into non-proliferating immature neutrophils (Gr1^+^, CD11b^+^, CD101^−^, Ly6G^int^, CXCR4^lo^, CXCR2^−^) and mature neutrophils (Gr1^+^, CD11b^+^, CD101^+^, Ly6G^+^, CXCR4^−^, CXCR2^+^), have been identified in mouse BM [2,13]. Further studies identified early committed progenitors of preNeu, designated as proNeu1 and proNeu2 [14]. The subset proNeu1, which is characterized in mouse BM as Lin^−^, CD115^−^, Flt3^−^, Ly6C^+^, CD117^hi^, CD34^+^, CD16/32^+^, CD106^−^ and CD11b^lo^, possesses self-renewing properties and differentiates into proNeu2, characterized by the enrichment of CD106^+^ and CD11b^hi^, compared to proNeu1 [14]. Importantly, proNeu1 and proNeu2 subsets have been identified in human cord blood [14]. These progenitors had the capacity to generate mature neutrophils after 3 days of in vitro culture [14].

The expression of transcription factors is required for the formation of mature neutrophils [2,15,16,17,18]. For instance, deficiency in specific transcription factors of the CCAAT/enhancer-binding protein (C/EBP) family leads to a disruption in neutrophil development [2,19,20]. C/EBPε is critical for proNeu2 development, while C/EBPα acts at the GMP stage, and their disruption in mice blocks the formation of neutrophils [14,19]. Other transcription factors, such as C/EBPβ and growth factor-independent-1 (Gfi-1) have also been involved in the process of neutrophil development [21,22].

### 2.1. Roles and Heterogeneity of Neutrophils in Steady State

Neutrophils have long been viewed as fully differentiated effector cells with a primary role in eliminating invading pathogens and a limited role in steady state. However, a growing body of evidence has shown that neutrophils are a heterogeneous population of cells in mouse and human tissues which can play an important role in maintaining tissue homeostasis [2,23,24]. For instance, infiltration of neutrophils into naïve lungs influences the transcriptional program of the tissue, which becomes associated with carcinogenesis and migration of other cell types, such as circulating tumor cells [23].

The release of neutrophils from the BM into the peripheral blood is tightly regulated by circadian rhythms, and freshly released mature neutrophils present a distinct phenotype compared to immature neutrophils and aged neutrophils [25,26,27,28]. CXCR4 plays a central role in BM retention of immature cells and BM homing of aged neutrophils. Decreased expression of CXCR4 associated with increased expression of CXCR2 drives the egress of neutrophils from the BM into the bloodstream [29]. Fresh mature neutrophils are released during the night and the early morning [26]. In the circulation, neutrophils undergo diurnal phenotypic alterations referred to as the aging program, which is linked to circadian regulation of the neutrophil transcriptional program [30]. In healthy mice, the number of aged neutrophils peaks at zeitgeber time 5 (ZT5, 5h after lights on), and this phenomenon is found associated with an increased expression of the transcription factor BMAL1, a clock protein involved in the regulation of the circadian rhythm [25]. Mechanistically, BMAL1 controls the production of CXCL2, which in turn signals through CXCR2 to induce neutrophil aging [25]. Aged neutrophils are characterized by reduced expression of CD62L, increased expression of CXCR4, CD11b, and CD49d, and the presence of a hypersegmented nucleus. Increased expression of CXCR4 facilitates their homing into the BM for their elimination. In addition, the aging of neutrophils has been associated with the progressive loss of granule content and the reduction of their capacity to form neutrophil extracellular traps (NETs) [25,30]. This cell-intrinsic program referred to as neutrophil “disarming”, may protect tissues from excessive inflammation and vascular damage, as shown in a mouse model of endotoxin and antibody-induced acute lung inflammation (ALI) [30]. However, migration of neutrophils to tissues is favored by neutrophil aging, and their presence can protect from infections [25]. Other neutrophil activities, including the production of reactive oxygen species and the release of cytokines, were not altered by aging.

A recent study from the Immunological Genome Project (Immgen) consortium applied scRNA-seq in combination with RNA velocity to identify neutrophil populations in BM, blood, and spleen of mice [31]. The study revealed that neutrophil heterogeneity states can be projected onto a single continuum, referred to as “neutrotime”. This spectrum reflects the chronological development of neutrophils, ranging from immature neutrophils, mainly in the BM, to mature neutrophils, mainly in the blood and spleen [31]. Interestingly, scRNA-seq data from human BM neutrophils showed that the neutrotime signature can be detected in human neutrophils [31]. A restricted set of transcription factors whose expression varies with neutrotime has been identified. For instance, the expression of *Cepbe* was enriched in the early neutrotime, while the expressions of *Atf3*, *Klf2, Junb, Jund*, and *Cepbb* were enriched in later neutrotime [31].

A population of neutrophils characterized by the expression of a set of interferon-stimulated genes (ISG) has been observed in mice and humans [31,32]. This subset of neutrophils could represent a population primed to fight infections and has been observed in tumors [33]. This observation points to a possible modification of the neutrophil phenotype depending on their environment. Indeed, recent studies revealed that neutrophils have variable lifetimes in different tissues and can acquire tissue-restricted phenotypes and functional properties [23,34]. For example, splenic neutrophils showed high expression of CD74 and CR2, while expression of CD14 and IL1β was higher in pulmonary neutrophils [34]. In silico analyses of the transcriptional signatures of tissue neutrophils revealed that pathways classically attributed to neutrophils, including cell chemotaxis and immune response, were enriched in BM and blood neutrophils, while splenic neutrophils showed enrichment of a pathways associated with B cell homeostasis, and intestine and pulmonary neutrophils were associated with angiogenesis and neuronal development [34]. Previous reports showed that splenic neutrophils can provide helper signals to B cells, including through the production of a proliferation-inducing ligand (APRIL), IL-21, and B-cell activating factor (BAFF) [35]. A pro-angiogenic activity of lung neutrophils has been confirmed in mouse models that require increased angiogenesis, such as after radiation-induced genotoxic injury [34]. The presence and retention of neutrophils in the lungs involve the expression of CXCR4 by neutrophils and CXCL12 by the pulmonary endothelium [36]. In addition to the pro-angiogenic activity of lung neutrophils, this tissue can act as a reservoir for neutrophils ready to be released into the circulation [36,37]. Accordingly, treatment with CXCR4 antagonists induced the release of neutrophils into the circulation [36].

Neutrophils can be detected in other tissues, including adipose tissue, liver, lymph nodes, and skeletal muscle [23,38]. Liver-infiltrating neutrophils have been involved in lipid metabolism, and lymph nodes neutrophils express a high level of major histocompatibility complex II (MHCII), suggesting a role in CD4^+^ T cell activation. The role of neutrophils in other tissues remains largely unknown [23,38].

### 2.2. Roles and Heterogeneity of Neutrophils in Cancer

A large number of studies have shown that the expression of inflammatory chemokines (e.g., CXCL1, CXCL2, CXCL5, CXCL8), cytokines (e.g., IL-1β, IL-17, TNFα), or other molecules, such as the complement component C5a, G-CSF, GM-CSF, or tumor-derived oxysterols, which are involved in the formation, mobilization, and recruitment of neutrophils, was increased in patients and mice with cancer [39,40,41,42,43,44,45,46,47,48]. Accordingly, increased blood neutrophilia is observed in cancer, and TANs represent an important component of the TME in the primary tumor, premetastatic niche, and metastasis [4,49].

As mentioned above, neutrophils can play both protumor and antitumor activities in cancer [2,4]. Protumor activities of neutrophils have been associated with a direct crosstalk with circulating tumor cells [50] and the production of different factors by neutrophils, including reactive oxygen species (ROS) and other molecules that support DNA damage and genomic instability [51,52,53,54,55], cytokines, chemokines, growth factors [3,56,57], NETs that support the formation of metastasis and protect tumor cells from CD8^+^ T cells and NK cells [58,59,60,61,62,63,64,65], proangiogenic factors such as vascular endothelial growth factor (VEGF) and matrix metalloproteinase-9 (MMP-9) [9,66,67,68,69], and immunosuppressive mediators such as arginase 1 (Arg1), prostaglandins, and ligands of immune checkpoints [9,70,71,72]. In addition, neutrophil-derived granule proteins can also sustain tumor growth. For instance, neutrophil elastase (NE), stored into neutrophil azurophil granules, can induce cancer cells proliferation via the degradation of insulin receptor substrate 1 (IRS-1), an inhibitor of phosphoinositide 3-kinase (PI3K). As a result, PI3K interacted with the mitogen platelet-derived growth factor receptor (PDGFR), leading to cell proliferation [73,74].

In contrast, a growing body of evidence showed that neutrophils can also play antitumor activities. In particular, neutrophils can eliminate tumor cells through a direct cytotoxic activity mediated by the production of ROS, TNF-related apoptosis-inducing ligand (TRAIL), and nitric oxide (NO) [42,75,76,77], can act as antigen-presenting cells (APC) associated with the antitumor immune response [78,79], or can sustain type 1 polarization and antitumor activity of a subset of unconventional T cells [80]. NE has also been associated with the antitumor activity of neutrophils. Indeed, human, but not murine, neutrophils secreted catalytically active NE which can kill cancer cells through the liberation of the CD95 death domain [81].

Cancer has served as a paradigm for myeloid cell heterogeneity and plasticity, first for macrophages and now for neutrophils [82]. The first level of heterogeneity involved the maturation status of neutrophils in the circulation and tumor tissues of individuals with cancer [46,83,84]. The program of maturation of neutrophils was found to be profoundly altered in tumor-bearing mice [85,86]. In a mouse model of orthotopic pancreatic cancer, the presence of immature neutrophils defined as Ly6G^low/+^CXCR2^−^ CD101^−^ increased in tumor-bearing mice, and mice with higher tumor burden showed higher infiltration of immature neutrophils into the pancreas [13]. In patients with melanoma, a heterogeneous population of early unipotent neutrophil progenitors (defined as NeP), susceptible to represents a mixture of ProNeu1, ProNeu2, and preNeu (see above), has been observed in blood and tumor tissue and associated with tumor progression [12]. Mechanistically, the protumor activity of immature neutrophils has been attributed to their ability to suppress T cell activation and proliferation [87]. The immunosuppressive population of neutrophils has been referred to as granulocytic myeloid-derived suppressor cells (G-MDSCs). MDSCs represent a heterogeneous population of mostly immature myeloid cells related to neutrophils (G-MDSC) or monocytes (M-MDSC) and functionally characterized by their immunosuppressive activity [88]. A population of immature neutrophils characterized by the expression of the class E scavenger receptor Lectin-type oxidized LDL receptor-1 (LOX-1), associated with the upregulation of genes related to endoplasmic reticulum (ER) stress, has been defined as a specific population of human, but not murine, G-MDSCs [89,90]. Human G-MDSCs have also been described as CD15^+^ CD66b^+^ CD33^dim^ HLA-DR^−^ cells, but this phenotype is also observed in other neutrophil subsets [91,92]. Other molecules, such as CD117 [12,13,93] and PD-L1 [12,71,94,95,96], have been associated with protumor neutrophils. In addition to these molecular markers, neutrophils can be separated by density gradient centrifugation into low-density neutrophils (LDNs) and normal-density neutrophils (NDNs) or high-density neutrophils (HDNs) [4,80,97]. LDNs have been mainly associated with the immunosuppressive activity of neutrophils. However, it is important to note that mature neutrophils can include LDNs and that all immature neutrophils are not immunosuppressive for T cells, suggesting that in addition to their maturity status, other features can drive the immunosuppressive activity of neutrophils [84]. Recently, scRNA-seq of human and mouse TANs from lung cancer revealed conserved subpopulations of TANs between species. Indeed, human and mouse tumors contained five and six subsets of neutrophils, respectively, but three gene expression modules were conserved between human and murine TANs. These three modules consisted of neutrophils expressing classic neutrophil markers that progressed to neutrophils expressing inflammatory cytokines and to a small subset of neutrophils expressing type I interferon response genes [33].

Macrophages have served as a paradigm for the plasticity and heterogeneity of myeloid cells in cancer, with classically activated M1 macrophages and alternatively activated M2 macrophages exerting antitumor and protumor activities, respectively [82,98]. Mirroring the M1/M2 paradigm, a polarization of TANs towards antitumor N1 neutrophils and protumor N2 neutrophils has been proposed [9]. In particular, under the pressure of TGFβ present in the TME, neutrophils polarized into protumor N2 neutrophils, characterized by the production of proangiogenic factors and immunosuppressive activity through the secretion of Arg1 [9]. In contrast, following TGF-β blockade or administration of IFNβ, neutrophils polarized into antitumor N1 neutrophils, characterized by cytotoxic activity towards tumor cells, reduced expression of the proangiogenic factors VEGF and MMP-9, increased expression of T cell-attracting chemokines (e.g., CCL3, CXCL9, and CXCL10), and capacity to support CD8^+^ T cells activation [9,68,99]. The terms N1 and N2 should be used with caution, as they refer to the different extremities of a continuum of neutrophil polarization states. This classification probably represents an oversimplification of the heterogeneity and plasticity of immature and mature neutrophils in cancer.

## 3. Chemokines and Chemokine Receptors Acting on Neutrophils

The superfamily of chemokines is composed of small chemoattractant peptides (4kDa) that are classified into four subfamilies (C, CC, CXC, and CX_3_C) according to the position of the first two cysteines present in their sequence. Chemokines bind to seven-transmembrane domain G protein-coupled receptors, known as chemokine receptors [100].

Neutrophils express high levels of CXCR1 and CXCR2, two chemokine receptors that bind a subclass of CXC chemokines characterized by the presence of a glutamate–leucine–arginine (ELR) sequence in the N-terminus before the CXC motif. Human CXCR2 binds the ELR+ chemokines CXCL1, CXCL2, CXCL3, CXCL5, CXCL6, CXCL7, and CXCL8 [101], while human CXCR1 binds only CXCL6 and CXCL8 [102]. There are many differences between the human and the mouse system in terms of ELR+ chemokines. Indeed, murine ELR+CXC chemokines are fewer than the human ones, and there is no murine homolog of human CXCL8, while its analogs are murine CXCL1, CXCL2, and CXCL5. Finally, murine CXCR1 binds only CXCL5 and CXCL6 [103]. (Table 1)

Despite the high degree of sequence homology and partial ligand overlap, CXCR1 and CXCR2 have distinct and non-redundant roles. Both receptors activate the NF-κB signaling pathway that sustains inflammation [104]. CXCR2 activates also the PI3K/Akt and mitogen-activated protein kinase (MAPK)/p38 signaling pathways that promote cell migration and survival [105]. In addition, CXCR2 has been reported to induce NET release through Src, extracellular signal-regulated kinase (ERK), and p38/MAPK signaling pathways [106]. In vivo experiments with CXCR2 knockout mice demonstrated that this receptor is required for tissue neutrophil infiltration, activation, and NET formation [107,108]. In patients with chronic obstructive pulmonary disease (COPD), clinical trials are ongoing with CXCR2 inhibitors to reduce NET formation and improve lung function [109]. Regarding CXCR1, this receptor is selective for the activation of phospholipase D (PLD) that promotes ROS production, while it is dispensable for neutrophil extravasation [110]. The role of CXCR1 in bacterial killing was demonstrated by in vivo experiments of *Pseudomonas aeruginosa* and *Candida albicans* infection [111,112]. Furthermore, patients carrying the genetic variant CXCR1–T276 are more susceptible to bacterial infections because their neutrophils have decreased degranulation and less ability to kill fungi [113].

CXCR2 is expressed at different levels during neutrophil maturation. It is expressed by mature neutrophils both in BM and in blood and is downregulated in extravasated neutrophils. CXCL1 produced by mast cells and bound to endothelial cells through the atypical chemokine receptor 1 (ACKR1) can induce a premature CXCR2 downregulation in neutrophils that reverse-transmigrate in the vessels. This phenomenon is increased in aged individuals and is probably relevant for dysregulated systemic inflammation associated with aging [114]. Finally, CXCR2 activated in an autocrine manner by CXCL2, induces neutrophil aging [25]. Neutrophils also express the atypical receptor CCRL2, which is very similar to chemokine receptors but does not bind chemokines. CCRL2 forms dimers with CXCR2 and regulates CXCR2 membrane expression [115].

CXCR4 expression on neutrophils is complementary to the one of CXCR2. Immature neutrophils in the BM express high levels of CXCR4, which, by interacting with its ligand CXCL12, provides a retention signal for immature neutrophils and progenitors [116,117]. Accordingly, genetic deletion of CXCR4 in myeloid cells resulted in a reduction of BM neutrophils and an increase in circulating neutrophils [118]. On the other hand, WHIM (warts, hypogammaglobulinemia, infections, myelokathexis) patients carrying a gain-of-function mutation in CXCR4, have a strong reduction of circulating neutrophils [119]. In inflammatory conditions, CXCR4 is downregulated or cleaved, which favors the mobilization of BM neutrophils into the blood [120]. The membrane expression of CXCR4 is upregulated in aged neutrophils, promoting their homing back to the BM and their phagocytosis by macrophages [116,121,122].

In inflammatory conditions and after extravasation, neutrophils completely change their chemokine receptor repertoire. They downregulate CXCR2 levels and upregulate the inflammatory CC receptors CCR1, CCR2, and CCR5. CCR1 is upregulated by IFNγ and GM-CSF [123] and was found necessary for neutrophil recruitment in murine models of renal and lung infections [124,125]. CCR2 was previously thought to be relevant only for monocyte recruitment, but it is also important for neutrophil mobilization [126] and recruitment to metastatic sites [77,127]. In addition to their role in neutrophil mobilization and recruitment, these receptors activate neutrophil phagocytic activity and ROS production [128]. A subpopulation of CCR6+ and CCR7+ neutrophils with antigen presenting capacity has also been described [129,130].

### 3.1. Role of Chemokines in Neutrophil Activities in Cancer

Neutrophilia is observed in patients with cancer, and TANs represent an important component of the TME. High levels of circulating neutrophils, neutrophil-to-lymphocyte ratio, and TANs have been associated with poor prognosis in most solid tumors [3,131]. Neutrophils are recruited to tumors mainly by ELR+-CXC chemokines produced by tumor cells (e.g., CXCL8, CXCL5, and CXCL6) and by neutrophils themselves (e.g., CXCL2) or by other cells of the tumor stroma. In addition, CXCL8 and CXCL3 are targets of oncogenic KRAS signaling and are overexpressed in many tumors such as colon, lung, liver, prostate, ovarian carcinoma, and melanoma [132,133,134].

Inhibition of CXCR1 and/or CXCR2 by pharmacological or genetic approaches showed that limiting neutrophil infiltration resulted in reduced tumor growth in murine models of pancreatic ductal adenocarcinoma [135], colorectal cancer [133], lung adenocarcinoma [74], and rhabdomyosarcoma [136]. In these models, reduced tumor growth has been associated with an inhibition of angiogenesis and a promotion of T cell response against tumors. In addition, CXCR1 and CXCR2 play an important role in the release of NETs by TANs, which in turn shield tumor cells from CD8 T cell and NK cell cytotoxicity (Figure 1) [64,137].

Contrasting results were reported regarding the role of CXCR2 in breast cancer. Inhibition of CXCR2 in breast carcinoma models reduced the recruitment of neutrophils into the tumor mass and increased the efficacy of chemotherapy [138,139]. On the other hand, genetic deletion of CXCR2 in the PyMT (polyoma middle T oncogene) model of breast cancer resulted in increased infiltration of TANs and promotion of tumor growth [140].

The role of CXCR2 and its ligands in neutrophil activation and maturation in the tumor context is still discussed. CXCL1 induced neutrophil transition from NDN to a low-density state (LD-NDN) with a phenotype similar to that of LDN, which could promote tumor development [141]. In addition, combined inhibition of CXCR2 and SHP2 in non-small cell lung cancer (NSCLC) models, selectively targeted a population of TANs with an immunosuppressive phenotype [142]. In accordance with these results, CXCR2 inhibition enhanced the therapeutic effect of cisplatin in an in vivo model of lung tumor, and CXCR2 expression in lung tumor patients is correlated with poor prognosis [143]. However, it was reported that CXCR2^−/−^ TANs have reduced capacity to kill tumor cells and increased production of angiogenic factors [140]. Accordingly, in patients with triple-negative breast cancer, a low level of CXCR2 was associated with a poor prognosis [144].

Opposing results were also found for the role of CXCR2 in metastasis. CXCR2 and its ligands were described as having an antimetastatic role in renal cell carcinoma where CXCL5 and CXCL8 produced by tumor cells can attract neutrophils with antitumor activity [145]. In an orthotopic mouse model of pancreatic ductal adenocarcinoma (PDAC), genetic deletion of CXCR2 in the host resulted in enhanced liver metastasis associated with the expansion of neutrophils and immature myeloid precursor cells [146]. In contrast, in a genetically engineered mouse model of PDAC, genetic and pharmacological inhibition of CXCR2 reduced metastasis formation and enhanced chemotherapy efficacy [147]. These contrasting results could be explained by at least two mechanisms: first, inhibitors target both CXCR1 and CXCR2, and the differential role of these receptors in tumor biology has not yet been demonstrated. Second, CXCR2 is not selectively expressed by neutrophils but is also expressed by cancer cells and by other myeloid cells, such as macrophages. Indeed, in a prostate cancer model, inhibition of CXCR2 resulted in decreased tumor growth due to the reeducation of tumor-associated macrophages [148].

CXCR4 is highly expressed in tumors by many cell types, and several clinical trials with CXCR4 inhibitors are ongoing in cancer patients. However, the specific role played by CXCR4 in neutrophil function in cancer has been investigated in a limited number of studies. In murine models of hepatocellular carcinoma, the CXCR4 antagonist AMD3100 reduced the immunosuppressive activity of myeloid cells and increased the efficacy of the antitumor immune response [149]. Selective deletion of CXCR4 in myeloid cells reduced melanoma and breast cancer lung metastasis. Protection was mediated by an increased release of BM neutrophils producing high levels of IL-18, which enhanced NK cells antitumor activities [150]. In addition, CXCR4 inhibition could impact on the migration to BM of aged neutrophils, which have been reported to sustain angiogenesis, tumor progression, and metastasis [151].

Neutrophils can also be mobilized and recruited at the tumor site through the CCL2–CCR2 axis. In contrast to monocytes, neutrophils recruited through this axis displayed anti-tumor activity due to their CCR2-dependent ability to produce oxygen radicals that kill tumor cells [77,127,152]. The expression of CC chemokine receptors in neutrophil progenitors is regulated by the atypical chemokine receptor ACKR2 expressed by hematopoietic progenitors [77].

### 3.2. Targeting Chemokines and Chemokine Receptors Expressed by Neutrophils to Promote Immunotherapy

Cancer immunotherapy represents an important strategy to treat cancer patients. In particular, the use of immune checkpoint inhibitors (ICIs) has improved the survival of cancer patients, including, among others, patients with melanoma, lung cancer, colorectal cancer (CRC), and urothelial carcinoma [153]. However, only a subset of patients responds to these therapies, and it is necessary to identify immunosuppressive mechanisms present in the TME that may contribute to hindering ICI efficacy. Preclinical and clinical data have suggested a role for neutrophils in these resistance mechanisms [154]. In addition, it is emerging that interfering with the chemokine system can improve the efficiency of immunotherapy through the reduction of the immunosuppression and the promotion of antitumor immune responses (Table 2).

The efficacy of anti-PD-1 treatment was improved by genetic deficiency or inhibition of CXCR1/2 in murine models of oral and lung carcinomas, rhabdomyosarcomas [136], pancreatic adenocarcinoma [147], and prostate cancer [155]. In a mouse model of colorectal cancer, resistance to anti-PD-1 therapy observed in KRAS^G12D^-expressing colon tumors could be overcome by inhibition of CXCR2 [133]. In a murine model of breast carcinoma, CXCR1/2 inhibition enhanced the efficacy of a dual blocker of PD-L1 and TGF-β [139]. In all these models, CXCR2 inhibition resulted in reduced TAN infiltration, accompanied by a concomitant increase of CD8 T cells. In addition, in a mouse model of oral cancer, blocking CXCR1 and CXCR2 with a dual inhibitor improved NK cell-based immunotherapy [156].

Prompted by these promising results obtained in preclinical models, phase I and II clinical trials are evaluating the targeting of CXCR1 and CXCR2 in combination with anti-PD-1 in patients with metastatic melanoma, pancreatic ductal carcinoma, and Ras-mutated MSS metastatic colon carcinoma [157]. A phase II clinical trial (NCT04050462) is ongoing with an anti-CXCL8 antibody (BMS-986253) in combination with nivolumab (anti-PD-1 antibody) in patients with advanced hepatocellular carcinoma (HCC).

Other clinical trials are also ongoing in cancer patients with CXCR4. It appeared that blocking the CXCL12–CXCR4 axis promoted the release of neutrophils from the BM but reduced their capability to infiltrate tumors. Indeed, in models of ovarian cancer, blocking CXCR4 with the antagonist AMD3100 increased the efficacy of anti-PD-1 treatment by reducing neutrophil infiltration [158]. Accordingly, a phase II trial (NCT02826486) combining anti-PD-1 (pembrolizumab) with a CXCR4 inhibitor (BL-8040/BKT140) in pancreatic cancer patients resulted in reduced neutrophil infiltration and increased cytotoxic T cell activity [159].

Finally, clinical trials are also assessing the role of CC chemokine receptors in immunosuppression and cancer growth mediated by TANs [154]. However, CCR5 expression by TANs has been correlated with a better response to immunotherapy in bladder cancer patients [160]. In addition, neutrophils expressing CCR5 were reported to be able to stimulate T cells in early stages of human lung cancer [161].

## 4. Concluding Remarks

Neutrophils are an important component of the tumor microenvironment, and despite being associated in most tumors with a poor prognosis, their role in cancer must be reconsidered due to their plasticity and heterogeneity. Indeed, in cancer patients, the maturation program of neutrophils is profoundly altered, and progenitors and immature neutrophils are released in the circulation and infiltrate the tumor. In addition, depending on the cytokine milieu, both immature and mature neutrophils can be activated and acquire protumor or antitumor activities.

The interplay between the chemokine receptors CXCR4 and CXCR2 regulates the circadian release of neutrophils from the bone marrow and their return when they are aged. Genetical and pharmacological inhibition of these chemokine receptors results in decreased tumor growth associated with a decrease in neutrophil infiltration. Inhibition of CXCR2 can block neutrophil recruitment in the TME but also their protumor effector, such as the release of NETs, while inhibition of CXCR4 unleashes neutrophil IL-18 production, which stimulates the antitumor activity of NK cells. In addition, both preclinical and clinal trials indicated that targeting CXCR2 and CXCR4 increased the efficacy of immunotherapy by reducing tumor neutrophil infiltration and promoting adaptive immune responses. On the contrary, the role of CXCR2 in neutrophil recruitment and activation at metastatic site is still debated, and genetic models with specific neutrophil deletion are needed. In addition, neutrophils can be recruited at the tumor site by CC chemokines acting on CCR1, CCR2, and CCR5, but their targeting must be carefully considered because these chemokines can promote neutrophil antitumor activity.

In conclusion, recent findings have highlighted the possibility of targeting chemokine receptors expressed by neutrophils to improve immunotherapy efficiency. However, a better understanding of their effects on dynamic changes within multiple maturation and activation states of neutrophils is required to improve and identify the best therapeutic strategies.

## Figures and Tables

**Figure 1 cancers-14-00680-f001:**
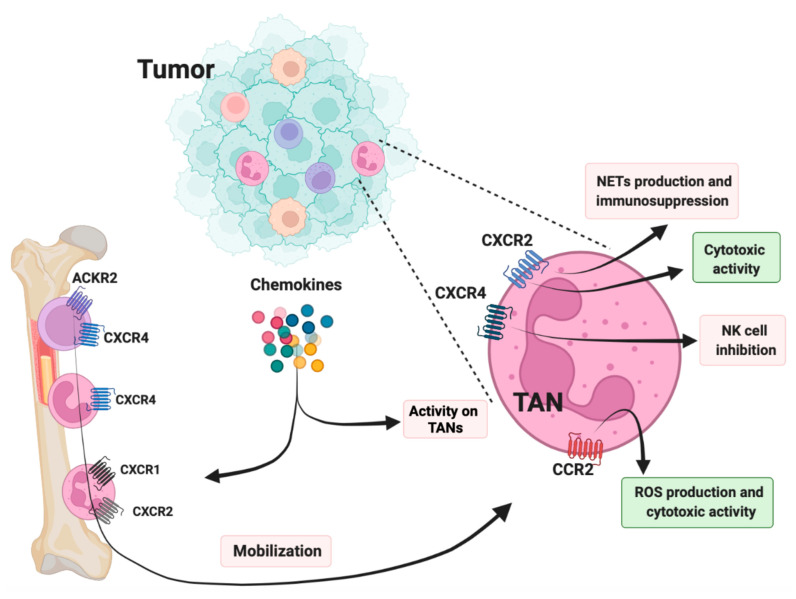
Role of chemokines and chemokine receptors in neutrophil recruitment and in pro- and antitumor activity of neutrophils. Hematopoietic stem and progenitor cells (HSPCs) and immature neutrophils are retained in the bone marrow by CXCR4. ACKR2 is expressed by HSPCs and inhibits neutrophil maturation and inflammatory CC chemokine receptor expression. Mature neutrophils upregulate CXCR1 and CXCR2 that promote neutrophil mobilization to the tumor site where ELR+ chemokines are overexpressed. Tumor-associated neutrophils (TAN) are endowed with protumor activities (red boxes) and antitumor activities (green boxes) that are induced by chemokine receptor signaling.

**Table 1 cancers-14-00680-t001:** Human and murine chemokine receptors expressed by neutrophils and their agonists and functions.

	CXCR1	CXCR2	CXCR4	CCR1	CCR2	ACKR2	CCRL2
Human endogenous agonists	CXCL6 CXCL8	CXCL1, 2, 3 CXCL5, 6, 7, 8	CXCL12	CCL3, 5 CCL7, 8 CCL13, 14, 15 CCL23	CCL2 CCL7 CCL13 CCL16	CCL2, 3, 4, 5 CCL7,8 CCL11 CCL13, 14 CCL17, 22	Chemerin
Murine endogenous agonists	CXCL5 CXCL6	CXCL1, 2, 3 CXCL5, 6	CXCL12	CCL3 CCL5,6,7 CCL9	CCL2 CCL7, CCL12	CCL2, 3, 4, 5 CCL7, 8 CCL11 CCL17, 22	Chemerin
Expression	Mature neutrophils (BM and blood)	Mature neutrophils (BM and blood)	Immature and aged neutrophils	Mature and extravasated neutrophils	Mature and extravasated neutrophils	Hematopoietic progenitors	Mature and activated neutrophils
Effector functions	ROS production	BM mobilization Extravasation NET production Aging	BM retention BM homing	Extravasation ROS production	BM mobilization ROS production	Regulation of CCR1,2, and 5 expression Checkpoint for neutrophil maturation	Regulation of CXCR2 expression

**Table 2 cancers-14-00680-t002:** Preclinical data on immunotherapy of tumors combined with chemokine receptor targeting.

Target	Inhibitor	Immunotherapy	Tumor Model	Effect	Refs
CXCR1 and CXCR2	CXCR1/2 inhibitor (SX-682)	Anti PD-1 mAb and adoptive T cell transfer	Oral and lung carcinoma	Decreased tumor growth and reduced immunosuppression	[162]
	Anti-mouse CXCR2 (MAB2164)	Anti PD-1 mAb	Rhabdomyosarcoma	Decreased tumor growth and reduced immunosuppression	[136]
	CXCR2 inhibitor (AZ13381758)	Anti PD-1 mAb	Pancreatic Ductal Adenocarcinoma	Decreased metastatization and reduced immunosuppression	[147]
	SX-682	Anti-CTLA4 and anti-PD1 mAbs	Metastatic castration-resistant prostate cancer	Reduced cancer growth and metastasis and increased T cell infiltrate	[155]
	SX682	Anti-PD1 mAb	Colorectal cancer	Reduced cancer growth and increased T cell infiltrate	[133]
	SX682	Anti-PD-L1 and TGFb	Breast cancer	Reduced tumor growth	[139]
	SX682	NK-cell-based immunotherapy	Head and neck carcinoma	Reduced tumor growth, reduced neutrophil infiltration, and enhanced KIL	[156]
CXCR4	AMD3100	Anti PD-1 mAb	Ovarian cancer	Reduced tumor growth and reduced neutrophil infiltration	[158]

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
