# Peer review of "Chemokines as Regulators of Neutrophils: Focus on Tumors, Therapeutic Targeting, and Immunotherapy"

_cancers, 2022, doi:10.3390/cancers14030680_

Round 1

Reviewer 1 Report

This is a review article with the aim to cover the role of neutrophils and the therapeutic potential of targeting its chemoattractants in solid tumors. The manuscript is well written and structured with good coverage of existing literature. However, it could benefit from the inclusion of more studies describing the mechanisms of tumor promotion or vice versa by neutrophils such as the production of neutrophil elastase (e.g. PMIDs: 20081861 & 24321240), or other neutrophil products in different types of cancers. The authors should also discuss more the potential mechanisms for improving immune checkpoint inhibitor efficacy by targeting neutrophils or its chemoattractant and related chemokine receptors. Most importantly, the authors have missed to include some of the important publications describing the importance and functional significance of CXCL1-2/CXCR1-2 axis as a target for treatment in cancer (e.g. PMIDs: 34680231, 34353854, 34070438, 33814009, 33000506, 32188703, 31852845, 24321240, 16618742), which could strengthen this review article. I’d also suggest to revise the illustration accordingly with more details.

Author Response

Point-by-point response to reviewer 1

  1. This is a review article with the aim to cover the role of neutrophils and the therapeutic potential of targeting its chemoattractants in solid tumors. The manuscript is well written and structured with good coverage of existing literature. However, it could benefit from the inclusion of more studies describing the mechanisms of tumor promotion or vice versa by neutrophils such as the production of neutrophil elastase (e.g. PMIDs: 20081861 & 24321240), or other neutrophil products in different types of cancers.

We thank the reviewer for his/her positive evaluation of our work.

The aim of our review was to discuss the activity of chemokines and chemokine receptors on neutrophils. Therefore, other more general mechanisms concerning the role played by neutrophils in cancer are mentioned and we have referred readers to the original articles and recent reviews concerning this aspect. However, we have included in the revised manuscript more studies concerning the role played by neutrophils in cancer, in particular, the activities of neutrophil Elastase are now discussed (see lines 190-204 and references 73,74 and 81).

  1. The authors should also discuss more the potential mechanisms for improving immune checkpoint inhibitor efficacy by targeting neutrophils or its chemoattractant and related chemokine receptors.Most importantly, the authors have missed to include some of the important publications describing the importance and functional significance of CXCL1-2/CXCR1-2 axis as a target for treatment in cancer (e.g. PMIDs: 34680231, 34353854, 34070438, 33814009, 33000506, 32188703, 31852845, 24321240, 16618742), which could strengthen this review article. I’d also suggest to revise the illustration accordingly with more details.

We thank the reviewer for this critical observation. We have included these references and expanded the discussion concerning this point in the revised manuscript (see lines 870-880 and references 141-145).

Reviewer 2 Report

This is a well-organized and comprehensive review of neutrophil development, tissue response, and impact on cancer from the viewpoint of chemokines. The following comments should be addressed prior to publication in Cancers.

Major comments:

  1. The sentence starting at line 351 inaccurately suggests that ICIs have improved survival for only the cancers listed. A more accurate statement could be: "In particular, the use of immune checkpoint inhibitors (ICIs) has improved the survival of several cancers including but not limited to patients with..."
  2. The sentence starting at line 354 needs stronger rationale. For example, "...and it is necessary to identify immunosuppressive mechanisms present in the TME that may contribute to the impediment of ICI efficacy.  

Minor comments:

A few typographical errors:

-line 85 should be released, not release

-line 123 the "b" should be changed to Greek

-line 130 Il-21 should be IL-21

-line 154 damages should be damage

-line 236 trough should be through

-line 305 delete the duplicate word associated

Author Response

Point-by-point response to reviewer 2

This is a well-organized and comprehensive review of neutrophil development, tissue response, and impact on cancer from the viewpoint of chemokines. The following comments should be addressed prior to publication in Cancers.

We thank the Reviewer for his/her assessment of our work and for the positive comments.

Major comments:

  1. The sentence starting at line 351 inaccurately suggests that ICIs have improved survival for only the cancers listed. A more accurate statement could be: "In particular, the use of immune checkpoint inhibitors (ICIs) has improved the survival of several cancers including but not limited to patients with..."
  2. The sentence starting at line 354 needs stronger rationale. For example, "...and it is necessary to identify immunosuppressive mechanisms present in the TME that may contribute to the impediment of ICI efficacy.  

We have taken into account the Reviewer's comments and modified the text accordingly.

Minor comments:

A few typographical errors:

-line 85 should be released, not release

-line 123 the "b" should be changed to Greek

-line 130 Il-21 should be IL-21

-line 154 damages should be damage

-line 236 trough should be through

-line 305 delete the duplicate word associated

We thank the reviewer for his/her careful revision of our review. We have modified the text according to his/her comments.